# Line strain representation and shear strain representation of 3D strain states

**Shunqun Li[1,2], Xuelei Cheng[3,4]\*, Jianbao Fu[5], Lin Pan[6], Ran Hai[3]**

**1** School of Civil Engineering, Tianjin Chengjian University, Tianjin, China, **2** State Key Laboratory of Geomechanics and Geotechnical Engineering, Institute of Rock and Soil Mechanics, Chinese Academy of Sciences, Wuhan, China, **3** School of Architecture and Civil Engineering, Zhongyuan University of Technology, Zhengzhou, Henan, China, **4** Department of Civil Engineering Mechanics, Yellow River Institute of Hydraulic Research, Zhengzhou, Henan, China, **5** Tianjin Port Engineering Institute Co. Ltd. of CCCC First Harbor Engineering, Tianjin, China, **6** School of Civil Engineering and Architecture, Xinxiang University, Xinxiang, Henan, China

\* xueleic@163.com

**Data Availability Statement:** All relevant data are within the paper.

**Funding:** This research was supported by the National Natural Science Foundation of China (41877251) and the Open Research Fund of State

## Abstract

The strain state in 3D space is usually expressed by the conventional method of combining three linear and shear strains. Due to the obvious differences between the first two strains, it is necessary to uncover their properties when describing deformation, studying yield and failure, and developing test apparatus or equipment. The difficulties encountered in the above work would be greatly simplified if strain states could be expressed in a single strain form, namely including only linear or shear strains. As a start, this paper explores the meaning and nature of strain states. Then, based on the hypothesis of small deformations, two strain state expressions, the linear strain expression method (LSEM) and shear strain expression method (SSEM), were established for incompressible materials with only linear strain and shear strain as parameters respectively. Furthermore, conditions, implementation steps and specific forms for the application of SSEM in 1D, 2D and 3D strain states are obtained. As an example, two representations based on tetragonal pyramid and rotating tetrahedron are especially given. Therefore, conventional strain representation methods can be expressed as a combination of line strains in a certain direction or a combination of characteristic shear strains. The results of this paper provide a new way for understanding deformation characteristics, revealing yielding process, establishing constitutive models, and developing testing apparatus or equipment.

## 1. Introduction

Stress and strain are two basic concepts of solid mechanics. A full and deep understanding on connotation and essence of these concepts is fundamental to any branch of mechanics [1, 2]. In fact, stress and strain are the inevitable and objective responses of the stressed body to external influences and do not depend on the will of the person However, the description of the reaction is subjective and artificial. As such, expressions of stress and strain can be diverse, variable, improvable, and updatable [3].

Key Laboratory of Geomechanics and Geotechnical Engineering (Z013002) through grants awarded to SL. The study also received funding from the Special Fund for Basic Scientific Research and Young Backbone Teachers of Zhongyuan University of Technology through grants awarded to XC (K2020QN015, 2020XQG14).

**Competing interests:** I have read the journal's policy and the authors of this manuscript have the following competing interests: [Geotechnical engineering]

Certain stress states for different mechanical problems often need to be described in different forms. For isotropic materials, a three-dimensional stress state containing three normal stresses and three shear stresses can be simplified to a principal stress state, namely $\{\sigma_1, \sigma_2, \sigma_3\}$. The stress state can be expressed in triaxial tests as first principal stress, third principal stress and medium principal stress coefficient, ie, $\{\sigma_1, \sigma_3, b\}$, in the triaxial test, as well as $\{I_1, J_2, \theta_\sigma\}$, $\{p, q, \theta_\sigma\}$ or $\{\sigma_{oct}, \tau_{oct}, \theta_\sigma\}$ and other forms in the study on constitutive model [4–6]. Describing the same event from different angles can greatly deepen people's understanding. For example, the description method of $\{p, q, \theta_\sigma\}$ can account for mechanism where volume changes and deformation changes come only from $p$ and $q$ respectively. Thus, some elastic, elasto-plastic or plastic models can be established based on the hypothesis. Another example is that a three-dimensional stress state can be expressed as six normal stresses in six particular directions, and a three-dimensional earth pressure chamber can be invented to test the complete stress state of the soil [7–9].

Based on the concept of shear strain, namely right angle deformation, the relationship between shear strain and three-dimensional strain was studied, with a strain state description method obtained. By this method, the strain state at a certain point can be equated to the combination of several characteristic shear strains.

## 2. Conventional representation of strain state

Line strain is the ratio of the change in length to the initial length of an object when it is deformed in a certain direction [10]. Shear strain refers to the change in right angles, expressed in radians, which is also called angular strain, shearing strain or relative shear deformation [11].

Depending on the shape, material properties and force qualities of the object, there are three forms of strain states at a point in the continuum, known as one-dimensional strain state, two-dimensional strain state and three-dimensional strain state [12], as shown in Fig 1.

The one-dimensional strain state is usually denoted as $\{\varepsilon_{yy}\}$. The two-dimensional strain state or the surface strain state is usually expressed as $\{\varepsilon_{xx}, \varepsilon_{yy}, \gamma_{xy}\}^T$. And the three-dimensional strain state is usually expressed as

$$\boldsymbol{\varepsilon}_{ij} = \{\varepsilon_{xx} \quad \varepsilon_{yy} \quad \varepsilon_{zz} \quad \gamma_{xy} \quad \gamma_{yz} \quad \gamma_{zx}\}^T \tag{1}$$

where, $ij$ represents $xx$, $yy$, $zz$, $xy$, $yz$, $zx$ in that order. Generally speaking, $\varepsilon_{xx}$, $\varepsilon_{yy}$, and $\varepsilon_{zz}$ are the line strains in the $x$, $y$, and $z$ directions, respectively; $\gamma_{xy}$, $\gamma_{yz}$, and $\gamma_{zx}$ are the corresponding three shear strains, respectively. These findings show that the strain state in three dimensional spaces is often expressed as the strain of a cubic element, including three line strains and three shear strains [13, 14].

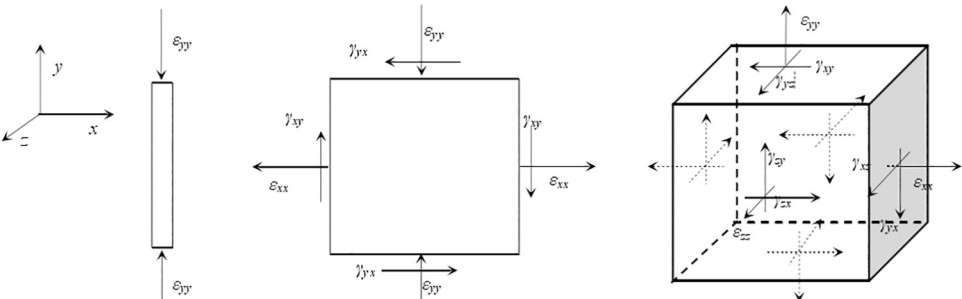

**Fig 1. Conventional representation for strain state of a continuous medium.** (a)One-dimensional strain. (b) Two-dimensional strain. (c) Three-dimensional strain.

For different engineering problems, strains can be expressed in different ways. In a three-dimensional strain space, the strain state at point $o$ can be write as $\varepsilon = \{\varepsilon_{xx}, \varepsilon_{yy}, \varepsilon_{zz}, \gamma_{xy}, \gamma_{yz}, \gamma_{zx}\}$ [15], as shown in Fig 1(C). If line $oA$ has two inclinations, as shown in Fig 2, the unit direction vector of $oA$, $\{l, m, n\}$, can be written as

$$l = \sin\delta\cos\varphi \tag{2}$$

$$m = \sin\delta\sin\varphi \tag{3}$$

$$n = \cos\delta \tag{4}$$

In Fig 2, Point $B$, angle $\delta$, and angle $\varphi$ are the projection of point $A$ on plane $xoy$, the angle between $oA$ and $z$-axis, and the angle between $oB$ and $x$-axis, respectively. Therefore, the line strain $\varepsilon_1$ along the $oA$ direction is

$$\varepsilon_1 = l^2\varepsilon_{xx} + m^2\varepsilon_{yy} + n^2\varepsilon_{zz} + 2lm\gamma_{xy} + 2mn\gamma_{yz} + 2nl\gamma_{zx} \tag{5}$$

Therefore, if the strain state of a point is known, then the line strain at that point in any direction can be obtained from Eq (5). Conversely, if the line strain at a point in 6 different directions is known, then six different linear equations for the unknown $\varepsilon_{xx}, \varepsilon_{yy}, \varepsilon_{zz}, \gamma_{xy}, \gamma_{yz}$, and $\gamma_{zx}$ can be obtained from Eq (5). According to these six linear equations and the relevant theories of linear algebra, the strain state represented by Eq (1) can be obtained.

Let the six different directions of linear strain be $\varepsilon_i$ ($i$ = 1, 2, 3, 4, 5, 6), then Eq (6) can be got according to Eq (5)

$$\varepsilon_i = l_i^2\varepsilon_{xx} + m_i^2\varepsilon_{yy} + n_i^2\varepsilon_{zz} + 2l_im_i\gamma_{xy} + 2m_in_i\gamma_{yz} + 2n_il_i\gamma_{zx} \tag{6}$$

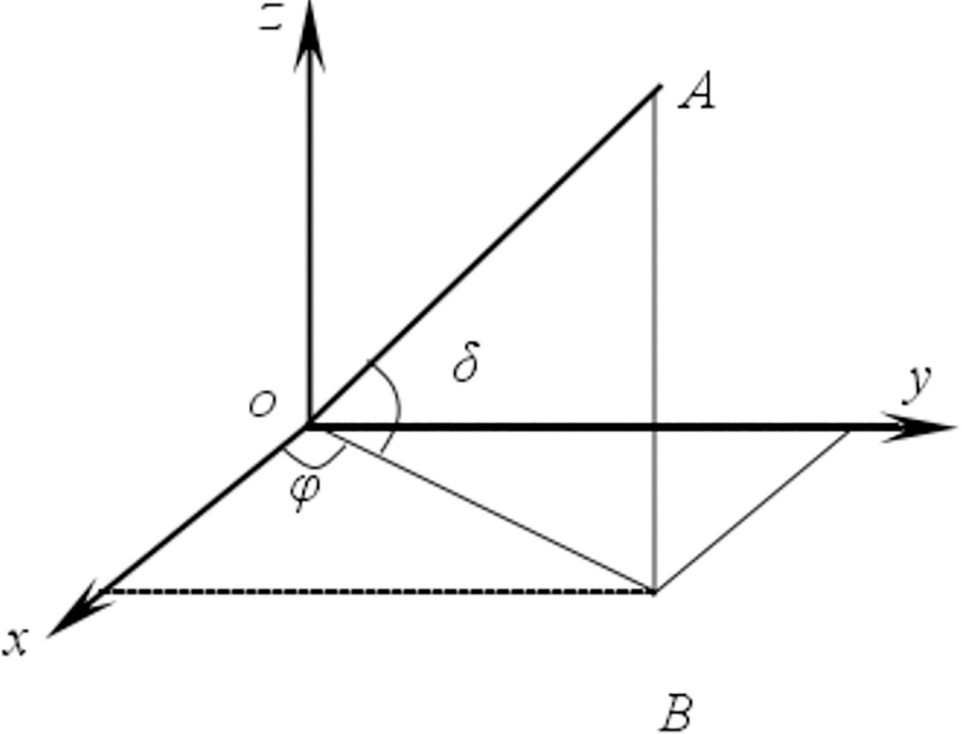

**Fig 2. Direction vector of space line.**

where, $l_i$, $m_i$, and $n_i$ are the unit direction vectors of the $i$-th line strain, respectively. After Eq (6) is expanded, Eq (5) will be received.

$$\begin{Bmatrix} \varepsilon_1 \\ \varepsilon_2 \\ \varepsilon_3 \\ \varepsilon_4 \\ \varepsilon_5 \\ \varepsilon_6 \end{Bmatrix} = \begin{Bmatrix} l_1^2 & m_1^2 & n_1^2 & 2l_1m_1 & 2m_1n_1 & 2n_1l_1 \\ l_2^2 & m_2^2 & n_2^2 & 2l_2m_2 & 2m_2n_2 & 2n_2l_2 \\ l_3^2 & m_3^2 & n_3^2 & 2l_3m_3 & 2m_2n_2 & 2n_3l_3 \\ l_4^2 & m_4^2 & n_4^2 & 2l_4m_4 & 2m_4n_4 & 2n_4l_4 \\ l_5^2 & m_5^2 & n_5^2 & 2l_5m_5 & 2m_5n_5 & 2n_5l_5 \\ l_6^2 & m_6^2 & n_6^2 & 2l_6m_6 & 2m_6n_6 & 2n_6l_6 \end{Bmatrix} \begin{Bmatrix} \varepsilon_{xx} \\ \varepsilon_{yy} \\ \varepsilon_{zz} \\ \gamma_{xy} \\ \gamma_{yz} \\ \gamma_{zx} \end{Bmatrix} \tag{7}$$

if

$$\Gamma = \begin{Bmatrix} l_1^2 & m_1^2 & n_1^2 & 2l_1m_1 & 2m_1n_1 & 2n_1l_1 \\ l_2^2 & m_2^2 & n_2^2 & 2l_2m_2 & 2m_2n_2 & 2n_2l_2 \\ l_3^2 & m_3^2 & n_3^2 & 2l_3m_3 & 2m_2n_2 & 2n_3l_3 \\ l_4^2 & m_4^2 & n_4^2 & 2l_4m_4 & 2m_4n_4 & 2n_4l_4 \\ l_5^2 & m_5^2 & n_5^2 & 2l_5m_5 & 2m_5n_5 & 2n_5l_5 \\ l_6^2 & m_6^2 & n_6^2 & 2l_6m_6 & 2m_6n_6 & 2n_6l_6 \end{Bmatrix} \tag{8}$$

Then, Eq (7) can be written as

$$\{\varepsilon_1 \quad \varepsilon_2 \quad \varepsilon_3 \quad \varepsilon_4 \quad \varepsilon_5 \quad \varepsilon_6\}^T = \Gamma\{\varepsilon_{xx} \quad \varepsilon_{yy} \quad \varepsilon_{zz} \quad \gamma_{xy} \quad \gamma_{yz} \quad \gamma_{zx}\}^T \tag{9}$$

where, $\Gamma$ is the transformation matrix. If $\Gamma$ is reversible, then

$$\{\varepsilon_{xx} \quad \varepsilon_{yy} \quad \varepsilon_{zz} \quad \gamma_{xy} \quad \gamma_{yz} \quad \gamma_{zx}\}^T = \Gamma^{-1}\{\varepsilon_1 \quad \varepsilon_2 \quad \varepsilon_3 \quad \varepsilon_4 \quad \varepsilon_5 \quad \varepsilon_6\}^T \tag{10}$$

A necessary and sufficient condition for the integrability of the matrix $T$ is that the matrix is either full rank or non-singular. Therefore, the three-dimensional strain state can be determined from the six-line strain, provided that the direction of the six-line strain is reasonably set to satisfy the conditions for the reversibility of the matrix $T$. That is to say, the three-dimensional strain state can be expressed by six line strains as Eq (7) is reversed.

The plane strain or the two-dimensional strain is usually expressed in the form shown in Fig 1(B). According to Eq (10), the plane strain can obviously be expressed in three line strains. In the plane strain problem, with $\delta = 90^\circ$, $l = \cos\varphi$, and $m = \sin\varphi$, Eq (8) is simplified to

$$\Gamma = \begin{Bmatrix} l_1^2 & m_1^2 & 2l_1m_1 \\ l_2^2 & m_2^2 & 2l_2m_2 \\ l_3^2 & m_3^2 & 2l_3m_3 \end{Bmatrix} \tag{11}$$

After obtaining the inverse matrix of Eq (11), the two-dimensional strain can be expressed by Eq (10). In engineering or experiment, the strain state at the measurement point can be easily obtained if the line strain in three directions is obtained, and this is how the general two-dimensional strain rosette work, as shown in Fig 3.

According to the linear strain representation method of the three-dimensional strain provided in Eq (10), a three-dimensional strain rosette device for three-dimensional strain testing is provided in the literature, including a regular tetrahedral shape and a one-point shape [16, 17], as shown in Fig 4.

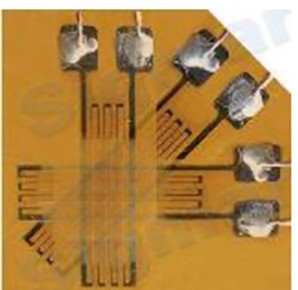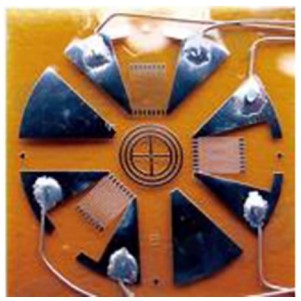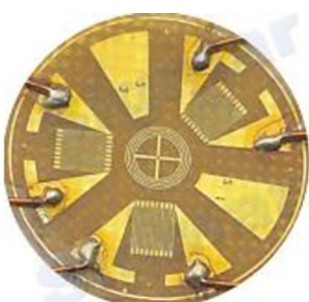

**Fig 3. Plane strain expressed by three line strains.**

The strain in the six special directions in Fig 4 can be measured by strain gauges. After calculating Eqs (2) through (4), Eq (8) and its inverse matrix can be obtained. The strain state of Eq (1) at the point of measurement can be obtained from Eq (10).

Fig 4 suggests that the line strain representation method is equivalent to the conventional strain state representation, as shown below. For two-dimensional strains, the line strains in three specific directions with strain gages in Fig 3 correspond to the strain states shown in Fig 1(B). For three-dimensional strain, the line strains in the six specific directions set by strain gages as shown in Fig 4 are also equivalent to strain states as shown in Fig 1(C). It can therefore be concluded that the strain state at one point can be expressed in line strain in multiple special directions.

## 3. Shear strain representation method for one-dimensional, two-dimensional strain state and three-dimensional strain state

Strain state can be expressed as combinations of line strains or shear strains. The relatively simple strain states, namely the one- and two-dimensional problems, are discussed here first, followed by the three-dimensional problems.

### 3.1 One-dimensional strain state

A one-dimensional strain state is also a strain state in which there is only one line strain. Assuming that the linear strain in $x$-axis direction be $\varepsilon_{xx}$, the variation at right angle $x'oy'$ will

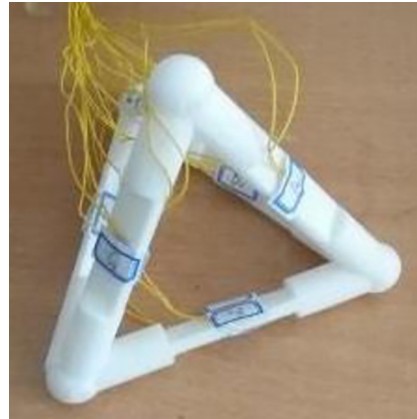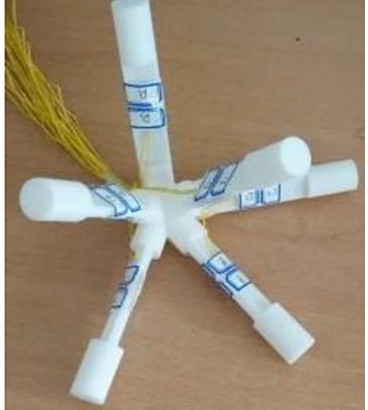

**Fig 4. Application of line strain representation method—three-dimensional strain rosette.** (a) Regular tetrahedron shape. (b) One point shape.

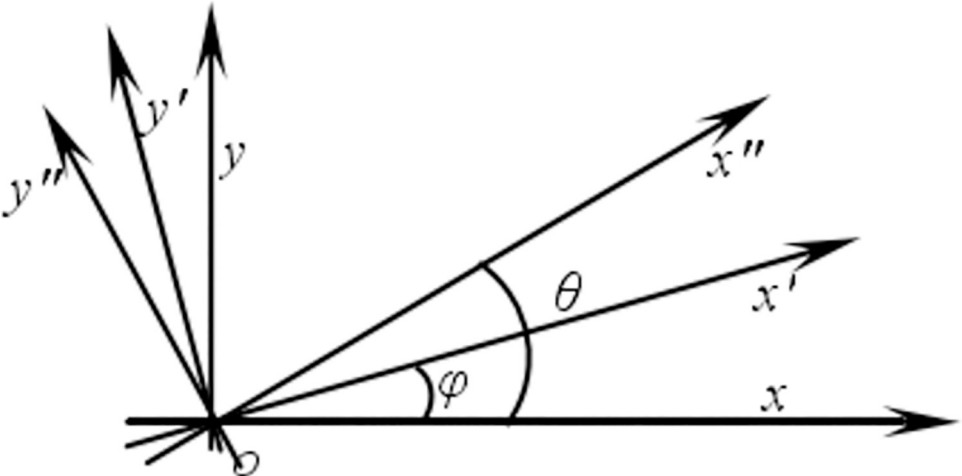

**Fig 5. Shear strain representation of simple strain state.**

be a shear strain $\gamma_\varphi$, as shown in Fig 5. That is

$$\gamma_\varphi = -\varepsilon_{xx}\sin2\varphi \tag{12}$$

If the angle $\varphi$ and the corresponding $\gamma_\varphi$ are known, then

$$\varepsilon_{xx} = -\gamma_\varphi\csc2\varphi \tag{13}$$

Fig 5 indicates that the one-dimensional line strain can be expressed in terms of shear strain. That is, if the shear strain $\gamma_\varphi$ based on the angle $\varphi$ is obtained by a certain method, the one-dimensional strain at the point can be obtained by Eq (13).

## 3.2 Two-dimensional strain state

By rotating the coordinate system $xoy$ around the origin $o$ by $\varphi$ and $\theta$, respectively, two new coordinate systems $x''oy''$ and $x'oy'$ can be obtained respectively, as shown in Fig 5. Then, the shear strain for the two-dimensional strain state ($\varepsilon_{xx}$, $\varepsilon_{yy}$, and $\gamma_{xy}$) in the new coordinate system is

$$\gamma_\varphi = -\varepsilon_{xx}\sin2\varphi + \varepsilon_{yy}\sin2\varphi + \gamma_{xy}\cos2\varphi \tag{14A}$$

$$\gamma_\theta = -\varepsilon_{xx}\sin2\theta + \varepsilon_{yy}\sin2\theta + \gamma_{xy}\cos2\theta \tag{14B}$$

If the material is incompressible, then

$$0 = \varepsilon_{xx} + \varepsilon_{yy} \tag{15}$$

Therefore

$$\begin{Bmatrix} \gamma_\varphi \\ \gamma_\theta \\ 0 \end{Bmatrix} = \begin{Bmatrix} -\sin2\varphi & \sin2\varphi & \cos2\varphi \\ -\sin2\theta & \sin2\theta & \cos2\theta \\ 1 & 1 & 0 \end{Bmatrix} \begin{Bmatrix} \varepsilon_{xx} \\ \varepsilon_{yy} \\ \gamma_{xy} \end{Bmatrix} \tag{16}$$

Let

$$\Gamma = \left\{ \begin{array}{ccc} -\sin2\varphi & \sin2\varphi & \cos2\varphi \\ -\sin2\theta & \sin2\theta & \cos2\theta \\ 1 & 1 & 0 \end{array} \right\} \tag{17}$$

If there exists an inverse of $\Gamma$, then $\Gamma^{-1}$ can be calculated. Eq (18) can be obtained from Eq (16) as bellow.

$$\left\{ \begin{array}{c} \varepsilon_{xx} \\ \varepsilon_{yy} \\ \gamma_{xy} \end{array} \right\} = \Gamma^{-1} \left\{ \begin{array}{c} \gamma_{\varphi} \\ \gamma_{\theta} \\ 0 \end{array} \right\} \tag{18}$$

Fig 1(B) shows that a two-dimensional strain state can be expressed by two shear strains corresponding to two different angles. That is, for incompressible materials, the representation of the two characteristic shear strains is equivalent to the representation of the strain state, as shown in Fig 1(B).

### 3.3 Shear strain representation method for three-dimensional strain state

According to the three-dimensional strain state shown in Fig 1(C), the change in the angle of inclusion of any two vertical vectors at this point can be obtained. Conversely, if the change in angle between a sufficient number of perpendicular vectors at a given point is known, then the strain state shown in Fig 1(C) can be obtained theoretically.

In three-dimensional space, the shear strain between any two mutually perpendicular directions $\alpha$ and $\beta$ is defined as $\gamma_{\alpha\beta}$, as shown in Fig 6, and then the relationship between $\gamma_{\alpha\beta}$ and the three-dimensional strain state shown in Fig 1(C) is

$$\gamma_{\alpha\beta} = 2(\varepsilon_{xx}a_1a_2 + \varepsilon_{yy}b_1b_2 + \varepsilon_{zz}c_1c_2) + \gamma_{xy}(a_1b_2 + a_2b_1) + \gamma_{yz}(b_1c_2 + b_2c_1) + \gamma_{zx}(c_1a_2 + c_2a_1) \tag{19}$$

where, $a_1$, $b_1$, $c_1$ and $a_2$, $b_2$, $c_2$ are the components of the two directions vector of $\alpha$ and $\beta$ on the three coordinate axes, respectively [18].

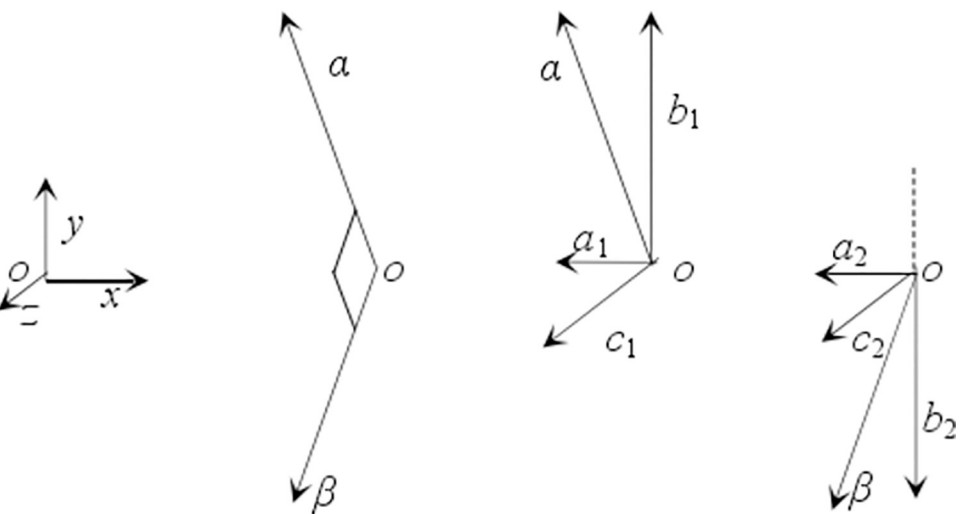

**Fig 6. Decomposition of two vertical direction vectors.**

To be simplified, six coefficients $\lambda_i$ are defined, i.e.

$$\lambda_1 = 2a_1 a_2 \tag{20A}$$

$$\lambda_2 = 2b_1 b_2 \tag{20B}$$

$$\lambda_3 = 2c_1 c_2 \tag{20C}$$

$$\lambda_4 = a_1 b_2 + a_2 b_1 \tag{20D}$$

$$\lambda_5 = b_1 c_2 + b_2 c_1 \tag{20E}$$

$$\lambda_6 = c_1 a_2 + c_2 a_1 \tag{20F}$$

Then, Eq (19) is rewritten as

$$\gamma_{\alpha\beta} = \varepsilon_{xx}\lambda_1 + \varepsilon_{yy}\lambda_2 + \varepsilon_{zz}\lambda_3 + \gamma_{xy}\lambda_4 + \gamma_{yz}\lambda_5 + \gamma_{zx}\lambda_6 \tag{21}$$

Thus, depending on the three-dimensional strain state, arbitrary right-angle variations can be obtained, that is

$$\gamma_{\alpha\beta i} = \varepsilon_{xx}\lambda_{i1} + \varepsilon_{yy}\lambda_{i2} + \varepsilon_{zz}\lambda_{i3} + \gamma_{xy}\lambda_{i4} + \gamma_{yz}\lambda_{i5} + \gamma_{zx}\lambda_{i6} \tag{22}$$

As a result, shear strains can be obtained for five or more azimuths, namely

$$\begin{Bmatrix} \gamma_{\alpha\beta1} \\ \gamma_{\alpha\beta2} \\ \gamma_{\alpha\beta3} \\ \gamma_{\alpha\beta4} \\ \gamma_{\alpha\beta5} \end{Bmatrix} = \begin{Bmatrix} \lambda_{11} & \lambda_{12} & \lambda_{13} & \lambda_{14} & \lambda_{15} & \lambda_{16} \\ \lambda_{21} & \lambda_{22} & \lambda_{23} & \lambda_{24} & \lambda_{25} & \lambda_{26} \\ \lambda_{31} & \lambda_{32} & \lambda_{33} & \lambda_{34} & \lambda_{35} & \lambda_{36} \\ \lambda_{41} & \lambda_{42} & \lambda_{43} & \lambda_{44} & \lambda_{45} & \lambda_{46} \\ \lambda_{51} & \lambda_{52} & \lambda_{53} & \lambda_{54} & \lambda_{55} & \lambda_{56} \end{Bmatrix} \begin{Bmatrix} \varepsilon_{xx} \\ \varepsilon_{yy} \\ \varepsilon_{zz} \\ \gamma_{xy} \\ \gamma_{yz} \\ \gamma_{zx} \end{Bmatrix} \tag{23}$$

For incompressible materials, the sum of three principal strains is equal to 0, i.e.

$$\varepsilon_{xx} + \varepsilon_{yy} + \varepsilon_{zz} = 0 \tag{24}$$

Eq (23) is combined with Eq (24) to obtain

$$\begin{Bmatrix} \gamma_{\alpha\beta1} \\ \gamma_{\alpha\beta2} \\ \gamma_{\alpha\beta3} \\ \gamma_{\alpha\beta4} \\ \gamma_{\alpha\beta5} \\ 0 \end{Bmatrix} = \begin{Bmatrix} \lambda_{11} & \lambda_{12} & \lambda_{13} & \lambda_{14} & \lambda_{15} & \lambda_{16} \\ \lambda_{21} & \lambda_{22} & \lambda_{23} & \lambda_{24} & \lambda_{25} & \lambda_{26} \\ \lambda_{31} & \lambda_{32} & \lambda_{33} & \lambda_{34} & \lambda_{35} & \lambda_{36} \\ \lambda_{41} & \lambda_{42} & \lambda_{43} & \lambda_{44} & \lambda_{45} & \lambda_{46} \\ \lambda_{51} & \lambda_{52} & \lambda_{53} & \lambda_{54} & \lambda_{55} & \lambda_{56} \\ 1 & 1 & 1 & 0 & 0 & 0 \end{Bmatrix} \begin{Bmatrix} \varepsilon_{xx} \\ \varepsilon_{yy} \\ \varepsilon_{zz} \\ \gamma_{xy} \\ \gamma_{yz} \\ \gamma_{zx} \end{Bmatrix} \tag{25}$$

Or abbreviated as

$$\boldsymbol{\gamma}_{\alpha\beta} = \boldsymbol{\lambda}\boldsymbol{\varepsilon} \tag{26}$$

Here

$$\boldsymbol{\gamma}_{\alpha\beta} = \left\{ \gamma_{\alpha\beta 1} \quad \gamma_{\alpha\beta 2} \quad \gamma_{\alpha\beta 3} \quad \gamma_{\alpha\beta 4} \quad \gamma_{\alpha\beta 5} \quad 0 \right\}^T \tag{27}$$

$$\boldsymbol{\varepsilon} = \left\{ \varepsilon_{xx} \quad \varepsilon_{yy} \quad \varepsilon_{zz} \quad \gamma_{xy} \quad \gamma_{yz} \quad \gamma_{zx} \right\}^T \tag{28}$$

$$\boldsymbol{\lambda} = \left\{ \begin{array}{cccccc} \lambda_{11} & \lambda_{12} & \lambda_{13} & \lambda_{14} & \lambda_{15} & \lambda_{16} \\ \lambda_{21} & \lambda_{22} & \lambda_{23} & \lambda_{24} & \lambda_{25} & \lambda_{26} \\ \lambda_{31} & \lambda_{32} & \lambda_{33} & \lambda_{34} & \lambda_{35} & \lambda_{36} \\ \lambda_{41} & \lambda_{42} & \lambda_{43} & \lambda_{44} & \lambda_{45} & \lambda_{46} \\ \lambda_{51} & \lambda_{52} & \lambda_{53} & \lambda_{54} & \lambda_{55} & \lambda_{56} \\ 1 & 1 & 1 & 0 & 0 & 0 \end{array} \right\} \tag{29}$$

If

$$R(\boldsymbol{\lambda}) = 6 \tag{30}$$

$\{\lambda\}^{-1}$ will exists. Eq (22) can then be further deduced as

$$\boldsymbol{\varepsilon} = \boldsymbol{\lambda}^{-1} \boldsymbol{\gamma}_{\alpha\beta} \tag{31}$$

According to these steps, the three-dimensional strain of the incompressible medium shown in Fig 1(C) can be expressed by five shear strains, provided that the matrix (30) corresponding to the five shear strains is invertible.

## 4. Case study on shear strain representation for strain state

In order to seek a feasible shear strain method for representing the three-dimensional strain state of incompressible materials, it is necessary to find five characteristic shear strains, namely five special right angles satisfying Eq (31).

### 4.1 Representation method based on quadrangular frustum

Fig 7 shows a geometric figure in which the upper and lower parts of a quadrangular pyramid are rectangular parallelepiped cut by the cubic element *ABCDEFGH*, here referred to as a quadrangular frustum. The quadrangular frustum can be obtained by the following process. First, the midpoints of three ridge lines *AE*, *BF*, and *EH* are determined. Then, the cubic elements are cut using the surface passing through the three midpoints. Thereby, a cutting surface $\eta_1$ will be obtained. Similarly, determining the midpoints of the three lines *DH*, *CG*, and *EH* gives a cutting surface $\eta_2$. Obviously, the surface $\eta_1$ is perpendicular to the surface $\eta_2$. In the same way, the cut surface $\xi_1$ and $\xi_2$ can also be accessed by cutting through the midpoints of the ridges *BF*, *CG*, *GH*, and *AE*, *DH*, *GH*. It is clear that the surface $\xi_1$ and the surface $\xi_2$ are also perpendicular to each other.

In addition, the surfaces *ABC*, *ABE*, and *AEH* are defined as $\pi$, $\chi$, and $\psi$, respectively. It is clear that any two of these three surfaces are also perpendicular to each other. The change in angle between surfaces $\pi$ and $\chi$ is therefore a shear strain, referred to here as $\gamma_{\pi\chi}$. Similarly, four shear strains of $\gamma_{\chi\psi}$, $\gamma_{\psi\pi}$, $\gamma_{\eta}$, and $\gamma_{\xi}$ can also be defined. The direction vectors corresponding to the above five shear strains are shown in Table 1.

In accordance with the definition of $\lambda_i$ in Eq (20), $\lambda_1$, $\lambda_2$ $\lambda_3$, $\lambda_4$, and $\lambda_5$ can be obtained based on the data in Tab. 1. Furthermore, according to Eq (30), a coefficient matrix based on the

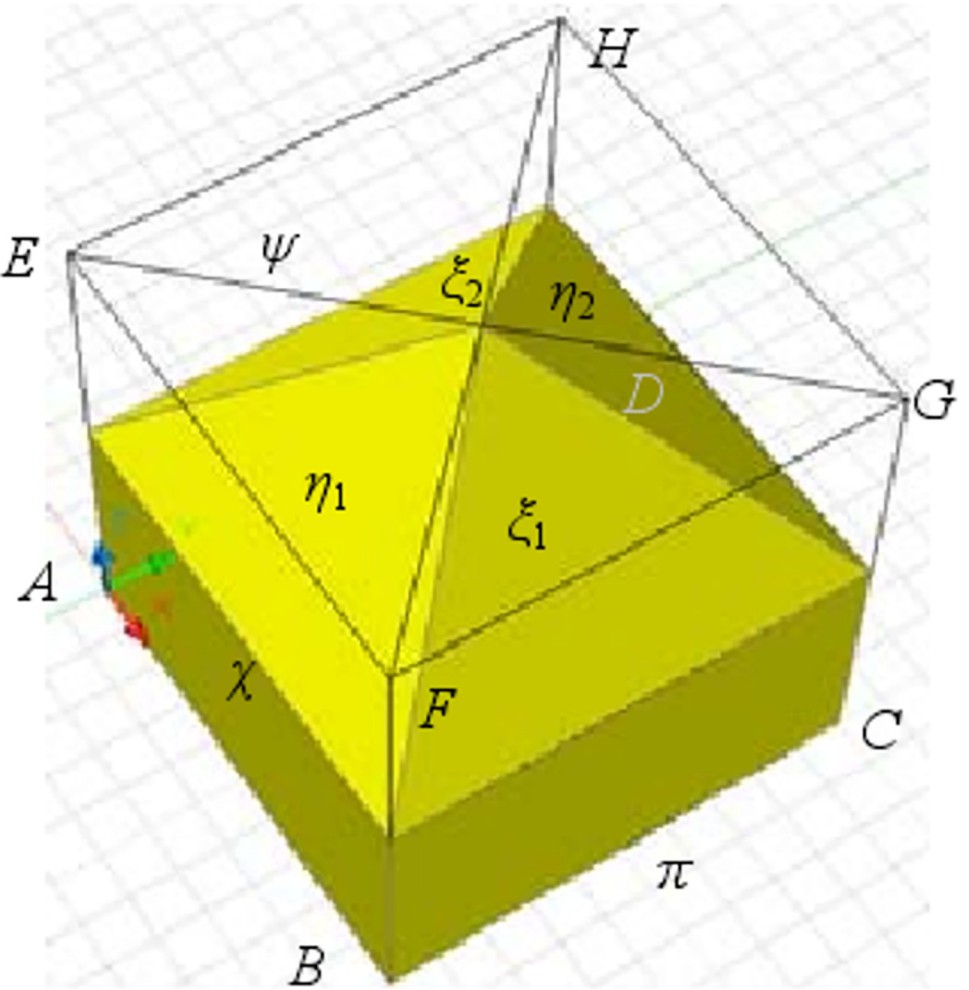

**Fig 7. Geometric representation of a quadrangular pyramid.**

shear strain representation method shown in Fig 7 can be achieved. That is

$$\lambda = \begin{Bmatrix} 0 & 0 & 0 & 0 & 1 & 0 \\ 0 & 0 & 0 & 1 & 0 & 0 \\ 0 & 0 & 0 & 0 & 0 & 1 \\ -1 & 0 & 1 & 0 & 0 & 0 \\ 0 & -1 & 1 & 0 & 0 & 0 \\ 1 & 1 & 1 & 0 & 0 & 0 \end{Bmatrix} \tag{32}$$

**Table 1. Direction vector of shear strain defined by a quadrangular pyramid.**

| Shear strain | Direction of $\alpha$ | | | Direction of $\beta$ | | |
|:---:|:---:|:---:|:---:|:---:|:---:|:---:|
| | $a_1$ | $b_1$ | $c_1$ | $a_2$ | $b_2$ | $c_2$ |
| $\gamma_{\pi\chi}$ | 0 | 0 | -1 | 0 | -1 | 0 |
| $\gamma_{\chi\psi}$ | 0 | -1 | 0 | -1 | 0 | 0 |
| $\gamma_{\psi\pi}$ | -1 | 0 | 0 | 0 | 0 | -1 |
| $\gamma_{\xi}$ | $\frac{\sqrt{2}}{2}$ | 0 | $\frac{\sqrt{2}}{2}$ | $-\frac{\sqrt{2}}{2}$ | 0 | $\frac{\sqrt{2}}{2}$ |
| $\gamma_{\eta}$ | 0 | $-\frac{\sqrt{2}}{2}$ | $\frac{\sqrt{2}}{2}$ | 0 | $\frac{\sqrt{2}}{2}$ | $\frac{\sqrt{2}}{2}$ |

The inverse of matrix (32) is

$$\lambda^{-1} = \begin{Bmatrix} 0 & 0 & 0 & -0.667 & 0.333 & 0.333 \\ 0 & 0 & 0 & 0.333 & -0.667 & 0.333 \\ 0 & 0 & 0 & 0.333 & 0.333 & 0.333 \\ 0 & 1 & 0 & 0 & 0 & 0 \\ 1 & 0 & 0 & 0 & 0 & 0 \\ 0 & 0 & 1 & 0 & 0 & 0 \end{Bmatrix} \tag{33}$$

Therefore, according to the shear strains in Eqs (33) and (31), the strain state of the incompressible medium shown in Fig 1(C) can be obtained if the changes in the five angles shown in Fig 7 are either known or can be measured.

## 4.2 Representation method based on rotating tetrahedron

The tetrahedral ABDE is obtained by cutting the cubic element ABCDEFGH with a three-dimensional surface as shown in Fig 8. After copying the tetrahedral *ABDE* and rotating it 60° along the straight line *AG*, another tetrahedron *AB'D'E'* will be obtained. The surfaces *AB'E'*, *AE'D'*, and *AD'B'* are defined as $\xi$, $\eta$, $\zeta$, respectively. Accordingly, the normal vectors of three surfaces $\xi$, $\eta$, and $\zeta$, are (0.667, 0.667, -0.333), (0.667, -0.333, 0.667) and (-0.333, 0.667, 0.667), respectively. Besides, the surfaces *ABC*, *ABE*, and *AEH* at the initial positions of the cube elements remain defined as $\pi$, $\chi$, and $\psi$, respectively.

Similarly to Eq (32), coefficient matrix corresponding to Fig 8 can be obtained. That is

$$\lambda = \begin{Bmatrix} 0 & 0 & 0 & 0 & 1 & 0 \\ 0 & 0 & 0 & 1 & 0 & 0 \\ 0 & 0 & 0 & 0 & 0 & 1 \\ 0.89 & -0.444 & -0.444 & 0.223 & 0.556 & 0.223 \\ -0.444 & -0.444 & 0.89 & 0.556 & 0.223 & 0.223 \\ 1 & 1 & 1 & 0 & 0 & 0 \end{Bmatrix} \tag{34}$$

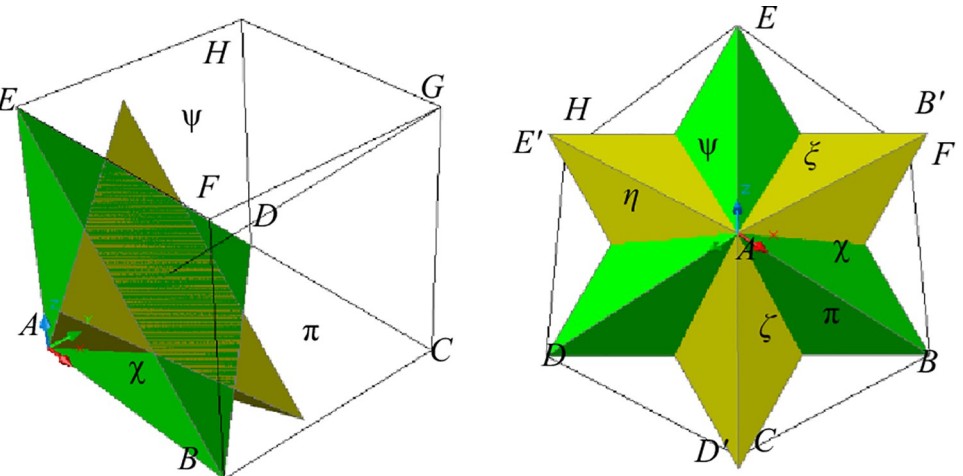

**Fig 8. Right-angled tetrahedron and rotated right-angled tetrahedron.** (a)Lateral view. (b)vertical view.

And the inverse matrix is

$$
\boldsymbol{\lambda}^{-1} = \left\{ \begin{array}{cccccc}
-0.417 & -0.167 & -0.167 & 0.75 & 0 & 0.333 \\
0.584 & 0.584 & 0.334 & -0.75 & -0.75 & 0.334 \\
-0.167 & -0.417 & -0.167 & 0 & 0.75 & 0.333 \\
0 & 1 & 0 & 0 & 0 & 0 \\
1 & 0 & 0 & 0 & 0 & 0 \\
0 & 0 & 1 & 0 & 0 & 0
\end{array} \right\}
\tag{35}
$$

Therefore, according to Eqs (35) and (31) and shear strains shown in Fig 8, the strain state of the incompressible medium in Fig 1(C) can also be obtained as described previously.

### 4.3 Error analysis

For incompressible materials, the strain state representation based on shear strain in Figs 7 and 8 are equivalent to those in Fig 1(C), that is, Eq (1) is equivalent to Eq (27). If the chance error for five shear strains in Figs 7 and 8 are $\Delta\gamma_0$, then the errors of each component in Eq (1) can be obtained respectively for Eq (33) and Eq (35).

$$
\Delta\varepsilon_{ij} = \Delta\gamma_0 \sqrt{\sum_{t=1}^{6} (\lambda_{jt}^{-1})^2}
\tag{36}
$$

Where, $\lambda_{jt}^{-1}$ is the $j$-th row and the $t$-th column of matrix $\boldsymbol{\lambda}^{-1}$. The error of the two representation methods is

$$
\{\Delta\boldsymbol{\varepsilon}_j\}^T = \Delta\gamma_0 \{0.817\ \ 0.817\ \ 0.577\ \ 1\ \ 1\ \ 1\}^T
\tag{37A}
$$

$$
\{\Delta\boldsymbol{\varepsilon}_j\}^T = \Delta\gamma_0 \{0.950\ \ 1.424\ \ 0.950\ \ 1\ \ 1\ \ 1\}^T
\tag{37B}
$$

It should be emphasized that the shear strain representation of a strain state is not unique, as is the linear strain representation. Theoretically, any set of shear strains satisfying Eq (31) can be expressed as Eq (1), which are equivalent to the strain state.

## 5. Conclusion

Deformation and strain are objective responses of human body to the external world, which are physical and non-anthropogenic. However, the description of material deformation is subjective and takes various forms. A class of representations of line and shear strains is developed based on the relationship between line, shear and conventional strain states. Then, the conditions and implementation steps of shear strain representation are obtained and studied. As two examples, representations based on quadrangular frustum and rotating tetrahedron are presented in detail, respectively. With the established methods, conventional strain representation is transformed into a linear strain combination or a characteristic shear strain combination in specific directions. As a result, a number of single strain representations, namely linear strain representation and shear strain representation, have been achieved. These findings are expected to provide some new means and ideas for the study failure, yielding and constitutive model.

## Author Contributions

**Conceptualization:** Xuelei Cheng.

**Data curation:** Lin Pan.

**Formal analysis:** Shunqun Li.

**Funding acquisition:** Xuelei Cheng.

**Investigation:** Lin Pan.

**Methodology:** Jianbao Fu.

**Project administration:** Shunqun Li, Xuelei Cheng.

**Resources:** Jianbao Fu.

**Supervision:** Xuelei Cheng.

**Validation:** Shunqun Li.

**Visualization:** Xuelei Cheng.

**Writing – original draft:** Shunqun Li.

**Writing – review & editing:** Ran Hai.

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
