## [Decision Letter · Decision Letter 0]

6 Jul 2021

PONE-D-21-09907

Line strain representation and shear strain representation of strain state

PLOS ONE

Dear Dr. Cheng,

Thank you for submitting your manuscript to PLOS ONE. After careful consideration, we feel that it has merit but does not fully meet PLOS ONE’s publication criteria as it currently stands. Therefore, we invite you to submit a revised version of the manuscript that addresses the points raised during the review process.

Please consider all the comments

We look forward to receiving your revised manuscript.

Kind regards,

Ahmed Mancy Mosa, Ph.D.

Academic Editor

PLOS ONE

Journal Requirements:

4. Please amend your authorship list in your manuscript file to include author Jianbao Fu.

Reviewers' comments:

Reviewer's Responses to Questions

**Comments to the Author**

1. Is the manuscript technically sound, and do the data support the conclusions?

Reviewer #1: Partly

Reviewer #2: Yes

2. Has the statistical analysis been performed appropriately and rigorously? 

Reviewer #1: Yes

Reviewer #2: N/A

3. Have the authors made all data underlying the findings in their manuscript fully available?

Reviewer #1: Yes

Reviewer #2: No

4. Is the manuscript presented in an intelligible fashion and written in standard English?

Reviewer #1: No

Reviewer #2: Yes

5. Review Comments to the Author

Reviewer #1: The paper aims to use linear strain expression method (LSEM) and shear strain expression method (SSEM) to represent the strain state in 3D space; however, major modifications should be done before publications as follows:

1. The title must be revised.

2. In the introduction, the references are old and need to be updated as well as previous studies on this work necessary to add in order to reflect the novelty of the paper.

3. Conventional representation of strain state is ambiguity and the structure of sentences need to be improved.

4. Please clarify this sentence “The strain in six special directions in Fig. 4 can be measured by strain gauges. After calculating equations (2) through (4), equation (8) and its inverse matrix can be obtained. Then, the strain state of equation (1) at the measurement point can be obtained according to equation (10)”.

5. The number of equation need to be checked “Therefore, if the strain state of a point is known, the line strain of the point in any direction can be obtained by equation (2.5)”. Similar mistakes should check through the manuscript.

6. What is the base of your assumption “Fig. 4 suggests that the line strain representation method is equivalent to the traditional strain state representation, as shown below”. The location of this sentence needs to be changed.

7. The format of Section 3 need to be changed and include the three dimensional state as a subtitle.

8. Please refer to the reference for this sentence ”The one-dimensional strain state is also the strain state with only one line strain”

9. The base of the derived equations such as equation 19 and 30 is weak and need to enhance by related references.

10. Why did the author choose the incompressible materials? Please clarify.

11. In the case study of Section 5, the comparison is not clear and also the author did not illustrate the reason about choosing the quadrangular frustum and rotating tetrahedron as case study.

12. The English language must be improved.

Reviewer #2: The authors explored about two strain state expression methods, called linear strain expression method (LSEM) and shear strain expression method (SSEM), for incompressible materials with only linear strain and shear strain as parameters respectively. In fact, to understand deformation characteristics, revealing yielding process, establishing constitutive models, and developing testing apparatus or equipment are interesting topic. However, the present study still needs some modifications to be more reliable for publication. The following are some comments that should be addressed before publishing:

1. It is preferring to stress out about the novelty of the topic in introduction section.

3. More newly studies could be cited in order to strengthen the literature review.

4. The conclusion needs to be explained more clearly about the objectives achieved.

6. PLOS authors have the option to publish the peer review history of their article (what does this mean?). If published, this will include your full peer review and any attached files.

Reviewer #1: **Yes: **Mohamed H. Mussa

Reviewer #2: No

---

## [Author Response · Author response to Decision Letter 0]

25 Aug 2021

Dear editor:

Thank you very much for your letter and advice. We have revised the manuscript, and would like to re-submit it for your consideration. We have addressed the comments raised by the reviewers, and the amendments are highlighted in red in the revised manuscript. Point by point responses to the reviewers’ comments are listed below this letter.

First of all, I am very grateful to the reviewers for their high evaluation of the work of this paper. We would continue to work hard to make the work more perfect in future.

Reviewer#1: The paper aims to use linear strain expression method (LSEM) and shear strain expression method (SSEM) to represent the strain state in 3D space; however, major modifications should be done before publications as follows:

Question 1：1. The title must be revised.

Answer 1: Already modified. I agree with the experts. The title ' Line strain representation and shear strain representation of strain state ' was modified as ' Line strain representation and shear strain representation for 3D strain state', and the whole paper was changed accordingly based on the modified title. The detailed modification has been marked in the red in the revised manuscript. Thanks again for the advice of the experts.

Question 2： In the introduction, the references are old and need to be updated as well as previous studies on this work necessary to add in order to reflect the novelty of the paper.

Answer 2: Already modified.

Question 3： Conventional representation of strain state is ambiguity and the structure of sentences need to be improved.

Answer 3: It is explained in Fig. 1.

Question 4： Please clarify this sentence “The strain in six special directions in Fig. 4 can be measured by strain gauges. After calculating equations (2) through (4), equation (8) and its inverse matrix can be obtained. Then, the strain state of equation (1) at the measurement point can be obtained according to equation (10)”.

Answer 4: The strain in six special directions in Fig. 4 can be measured by strain gauges because they are all line strain. After calculating equations (2) through (4), equation (8) and its inverse matrix can be obtained. Then, the strain state of equation (1) at the measurement point can be obtained according to equation (10).

Question 5： The number of equation need to be checked “Therefore, if the strain state of a point is known, the line strain of the point in any direction can be obtained by equation (2.5)”. Similar mistakes should check through the manuscript.

Answer 5: The number of equation has been checked through the manuscript. The detailed modification has been marked in the revised manuscript. Thanks again for the advice of the experts.

Question 6: What is the base of your assumption “Fig. 4 suggests that the line strain representation method is equivalent to the traditional strain state representation, as shown below”. The location of this sentence needs to be changed.

Answer 6: Because the six line strain and the traditional strain state can be displayed by each other.

Question 7: The format of Section 3 need to be changed and include the three dimensional state as a subtitle.

Answer 7: Already modified. I agree with the experts.

Question 8: Please refer to the reference for this sentence ”The one-dimensional strain state is also the strain state with only one line strain”

Answer 8: Already modified. I agree with the experts. The detailed modification has been marked in the revised manuscript. Thanks again for the advice of the experts.

Question 9: The base of the derived equations such as equation 19 and 30 is weak and need to enhance by related references.

Answer 9: Already modified. I agree with the experts. The detailed modification has been marked in the revised manuscript. While, Equation 30 is an assumption. 

Thanks again for the advice of the experts. 

Question 10: Why did the author choose the incompressible materials? 

Answer 10: Only the incompressible material (satisfying equation (24)) can satisfy equation (29). That is, compressible materials cannot satisfy equation (29). 

Question 11. In the case study of Section 5, the comparison is not clear and also the author did not illustrate the reason about choosing the quadrangular frustum and rotating tetrahedron as case study.

 Answer12: Already illustrated the reason in the revised manuscript.

Question 12: The English language must be improved.

Answer12: Already modified. I agree with the experts. The detailed modification has been marked in the revised manuscript. Thanks again for the advice of the experts.

Reviewer #2: The authors explored about two strain state expression methods, called linear strain expression method (LSEM) and shear strain expression method (SSEM), for incompressible materials with only linear strain and shear strain as parameters respectively. In fact, to understand deformation characteristics, revealing yielding process, establishing constitutive models, and developing testing apparatus or equipment are interesting topic. However, the present study still needs some modifications to be more reliable for publication. The following are some comments that should be addressed before publishing:

 Question 1. It is preferring to stress out about the novelty of the topic in introduction section.

Answer 1: Already modified. I agree with the experts. The novelty of the topic in introduction section has been stressed out. The detailed modification has been marked in the revised manuscript. Thanks again for the advice of the experts.

Question 2. More newly studies could be cited in order to strengthen the literature review.

Answer 2: The latest literature has been added. I agree with the experts. The detailed modification has been marked in the revised manuscript. Thanks again for the advice of the experts.

Question 3. The conclusion needs to be explained more clearly about the objectives achieved.

Answer 1: Already modified. I agree with the experts. The conclusion has been explained more clearly about the objectives achieved. The detailed modification has been marked in the revised manuscript. Thanks again for the advice of the experts.

We hope that the revised version of the manuscript is now acceptable for publication in your journal. If you have any queries, please don’t hesitate to contact me.

I look forward to hearing from you soon.

With best wishes,

Yours sincerely,

Xuelei Cheng

---

## [Decision Letter · Decision Letter 1]

25 Oct 2021

Line strain representation and shear strain representation of 3D strain state

PONE-D-21-09907R1

Dear Dr. Cheng,

We’re pleased to inform you that your manuscript has been judged scientifically suitable for publication and will be formally accepted for publication once it meets all outstanding technical requirements.

Kind regards,

Ahmed Mancy Mosa, Ph.D.

Academic Editor

PLOS ONE

Additional Editor Comments (optional):

Reviewers' comments:

Reviewer's Responses to Questions

**Comments to the Author**

1. If the authors have adequately addressed your comments raised in a previous round of review and you feel that this manuscript is now acceptable for publication, you may indicate that here to bypass the “Comments to the Author” section, enter your conflict of interest statement in the “Confidential to Editor” section, and submit your "Accept" recommendation.

Reviewer #2: All comments have been addressed

2. Is the manuscript technically sound, and do the data support the conclusions?

Reviewer #2: Partly

3. Has the statistical analysis been performed appropriately and rigorously? 

Reviewer #2: No

4. Have the authors made all data underlying the findings in their manuscript fully available?

Reviewer #2: Yes

5. Is the manuscript presented in an intelligible fashion and written in standard English?

Reviewer #2: Yes

6. Review Comments to the Author

Reviewer #2: The specific changes have been noted in the updated manuscript. congratulations on your accomplishments.

7. PLOS authors have the option to publish the peer review history of their article (what does this mean?). If published, this will include your full peer review and any attached files.

Reviewer #2: No

---

## [Editor Report · Acceptance letter]

10 Nov 2021

PONE-D-21-09907R1 

Line strain representation and shear strain representation of 3D strain states 

Dear Dr. cheng:

I'm pleased to inform you that your manuscript has been deemed suitable for publication in PLOS ONE. Congratulations! Your manuscript is now with our production department. 

Kind regards, 

on behalf of

Dr. Ahmed Mancy Mosa 

Academic Editor

PLOS ONE